# Constitutive Genetic Deletion of *Hcn1* Increases Alcohol Preference during Adolescence

**DOI:** 10.3390/brainsci10110763

**Published:** 2020-10-22

**Authors:** Michael C. Salling, Neil L. Harrison

**Affiliations:** 1Louisiana State University Health Sciences Center, New Orleans, LA 70112, USA; 2Columbia University Irving Medical Center, New York, NY 10032, USA; nh2298@cumc.columbia.edu

**Keywords:** *Hcn1*, alcohol, prefrontal cortex

## Abstract

The hyperpolarization-activated cyclic nucleotide-gated channel (HCN), which underlies the hyperpolarization-activated cation current (I_h_), has diverse roles in regulating neuronal excitability across cell types and brain regions. Recently, HCN channels have been implicated in preclinical models of substance abuse including alcohol. In the prefrontal cortex of rodents, HCN expression and I_h_ magnitude are developmentally regulated during adolescence and may be vulnerable to alcohol’s effects. In mice, binge alcohol consumption during the adolescent period results in a sustained reduction in I_h_ that coincides with increased alcohol consumption in adulthood, yet the direct role HCN channels have on alcohol consumption are unknown. Here, we show that the genetic deletion of *Hcn1* causes an increase in alcohol preference on intermittent 2-bottle choice task in homozygous null (HCN1*^−/−^*) male mice compared to wild-type littermates without affecting saccharine or quinine preference. The targeted viral deletion of HCN1 in pyramidal neurons of the medial prefrontal cortex resulted in a gradual loss of *Hcn1* expression and a reduction in I_h_ magnitude during adolescence, however, this did not significantly affect alcohol consumption or preference. We conclude that while HCN1 regulates alcohol preference, the genetic deletion of *Hcn1* in the medial prefrontal cortex does not appear to be the locus for this effect.

## 1. Introduction

Drinking excessive alcohol during adolescence is associated with cognitive dysfunction and an increased likelihood of developing an alcohol use disorder (AUD) later in life [1,2]. Within the adolescent period, several brain regions are still undergoing significant maturation which likely makes them more vulnerable to the pharmacological effects of high levels of alcohol consumption [3,4]. Neuroimaging studies have observed that heavy adolescent drinking accelerates the normal reductions in grey matter that are characteristic of adolescent brain development. These changes occur particularly in frontal, hippocampal, cerebellar, and limbic subregions, [5,6,7,8] which play central roles in reward seeking, executive function and associative learning. Measurements of brain activity, by functional magnetic resonance imaging or electroencephalogram recordings, indicate that adolescent drinkers display altered activity during response inhibition [9], spatial working memory [10,11] and alcohol cue reactivity [12] in frontal and limbic subregions. These findings demonstrate that the adolescent period is a critical node in the etiology of alcohol use disorders and that changes in neuronal function may be both a consequence and central regulator of addictive behaviors toward alcohol.

Preclinical research has shown that chronic alcohol exposure use can alter neuronal excitability [13,14]. In the rodent prefrontal cortex (PFC), both chronic intermittent alcohol vapor and voluntary alcohol consumption result in changes to both the synaptic and intrinsic activity of medial prefrontal neurons [15,16,17,18,19]. The adolescent period appears to have unique neuroadaptations to chronic alcohol use [20]. In a previous study, we found that intermittent alcohol consumption during early adolescence (postnatal day 30) to early adulthood (postnatal day 60) results in persistent alterations of neuronal excitability that are due to a reduction of the hyperpolarization-activated cation current (h-current or I_h_), and are associated with working memory deficits and changes in alcohol drinking typography [21]. 

The h-current is mediated by the family of hyperpolarization-activated cyclic nucleotide-gated cation channels (gene name *HCN*). This voltage-gated ion channel has unique properties conferred by its rather large voltage sensor that opens the channel during hyperpolarization [22]. The HCN channel pore is selective to both sodium and potassium cations at a 3:1 ratio, causing membrane depolarization. The HCN channel family is composed of *Hcn1*, *2*, *3* and *4* genes with *Hcn2* and *Hcn4* being highly expressed in the heart and biased to more subcortical regions of the brain and *Hcn1* having higher expression in the cortex and hippocampus [23]. In the sinoatrial node of the heart, HCN channels are responsible for pacemaker activity. In the brain, HCN channels have diverse actions. For example, they are important for establishing rhythmic firing in the thalamus and ventral tegmental area, while in the cortex they contribute to resting membrane potential, influence somatic and dendritic input resistance, and synaptic integration [24]. Intriguingly, in rodents, the HCN1 subunit channel undergoes developmental regulation during adolescence, which shows showing increased expression and function in pyramidal neurons in the hippocampus and prefrontal cortex as mice age from early adolescence to adulthood [21,25,26].

Genetic deletion experiments in mice have shown that HCN channels play unique roles in learning and memory. For instance, the deletion of HCN1 in the forebrain caused an enhancement of performance on a spatial memory task [27], while HCN1 deletion in the medial prefrontal cortex (PFC) negatively affected working memory performance through increased proactive interference [28]. While it is clear that HCN1 channels have a role in learning and memory, the role of these channels in drug intake has not yet been established. Based on our previous study, we hypothesized that the genetic deletion of *Hcn1* would result in increased alcohol consumption in mice and that the medial PFC would place a central role in this effect. To test this hypothesis, we assessed voluntary intermittent alcohol consumption in both adolescent HCN1 knockout mice and mice with a viral deletion of *Hcn1* specifically in the medial PFC.

## 2. Materials and Methods

**Subjects:** Constitutive HCN1 knockout (KO) [29] and conditional floxed HCN1 (HCN1*^f/f^*) male mice [27] were used in the study during the postnatal day (PND) 25–90. Both lines were backcrossed to C57BL/6J and 129s/SvEv backgrounds in order to produce the 50:50 offspring used in the study. Mice were weaned and transferred to their final colony room at PND 21–23, where they were initially group housed and then separated into individual cages at PND 25 prior to the initiation of the 2-bottle choice task at PND 30. Mice were given food and water ad libitum throughout experiment. Experimental procedures were performed under the guidelines set forth in the National Institute of Health Guide for the Care and Use of Laboratory Animals and with the approval of the Institutional Animal Care and Use Committee of Columbia University (code: AAAL4600). Female mice were not used in the current study.

**Two-bottle choice procedure**: Under this procedure, the mice were given 2 drinking bottles made from 50 mL Nalgene conical tubes (Thermofisher Scientific, Waltham, MA, USA) with double ball bearing sipper tubes placed inside bottle stoppers (Ancare, Bellmore, NY, USA). Mice were single housed and given 24 h intermittent access to one bottle of water and one bottle of 15% ethanol (*v/v*) every other day referred to here as intermittent access to alcohol (IA EtOH) from PND 30 to PND 60 with 2 water bottles on the alcohol free days [30]. Bottles and mice were weighed at approximately 1 h before the start of the dark cycle. A “drip cage” (mouse-free cage with bottles placed on rack) was used to measure the daily drip and evaporation of bottle solutions which were subtracted from each daily bottle weight value. Locations of bottles (right/left) were counterbalanced among mice and switched every two days to prevent side bias. Following 30 days of IA EtOH, HCN1*^+/+^*, HCN1^+/−^, and HCN1^−/−^ mice were given 2 weeks of water only and then given saccharine (Sigma Aldrich, St. Louis, MO, USA) at increasing doses (0.033%, 0.066% *w/v*) 2 days in a row at each concentration with the bottle side switched each day. Mice were then placed on water for 2 days and given increasing concentrations (30, 60 and 90 µM) of quinine hemisulfate (Sigma) diluted in water over 2 days per concentration with the bottle side switched each day. Mouse weight, fluid weight, bottle preference, and the alcohol dose consumed were calculated for each 2-bottle choice day. 

**Stereotaxic surgery:** At ~PND 25, the HCN1*^f/f^* mice were injected bilaterally with 200 nL adeno-associated virus serotype 2.8 (AAV8) carrying either Cre-recombinase fused to green fluorescent protein (CreGFP) or enhanced green fluorescent protein (EGFP) downstream from the calmodulin dependent kinase 2 (CaMK2a) promotor. AAV8-CaMK2a-CreGFP or AAV8-CaMK2a-EGFP (UNC Vector Core, Chapel Hill, NC, USA) was diluted to 2.0 × 10^11^ vg/mL and infused at 50 ul per minuteinto the prelimbic region of the medial prefrontal cortex at coordinates (in mm) AP: +1.7, M-L: ±0.35, D-V: −1.6 (from brain surface) using a mouse stereotax (Stoelting, Wood Dale, IL, USA). A subset of these mice were used to confirm the reduction of the HCN1 protein either using immunohistochemistry or brain slice electrophysiology. The remaining mice were used in the 2-bottle choice alcohol drinking experiment following recovery 4–7 days post injection (~PND 30). After 30 days of IA EtOH, their brains were collected to determine the viral injection target and mice with missing or mistargeted GFP expression were removed from the experiment (*n* = 3).

**Immunohistochemistry:** Mice were transcardially perfused with ice cold 0.1 M PBS and fixed with 4% paraformaldehyde (Electron Microscopy Sciences, Hatfield, PA, USA) and 50 µm brain slices were collected on a VT1200S vibratome (Leica Microsystems, Buffalo Grove, IL, USA). Free floating sections were blocked with 5% goat serum (Jackson Immunoresearch Laboratories) in PBS with 0.3% triton-x (PBST) and incubated overnight with the following primary antibodies: monoclonal anti-mouse HCN1 Igg1 1:200 (Neuromab, Davis, CA, USA), polyclonal chicken anti-GFP IgY 1:1000 (Abcam, Cambridge, UK), polyclonal rabbit anti-Cre IgG (gift from Christoph Kellendonk). On day 2, the sections were washed with PBST and incubated in the following secondary antibodies at 1:500 concentration: goat anti-chicken Alexa 488, goat anti-rabbit Alexa 647 and Alexa 546 goat anti-mouse IgG1 (Thermofisher). Sections were mounted onto slides, cover slipped with Diamond prolong antifade with 4’6-diamidino 2-phenylindole (DAPI) mounting medium and images were collected on an A1 confocal microscope (Nikon). Images were collected to qualitatively demonstrate *Hcn1* and Cre recombinase expression and to determine the placement of viral injections.

**Slice electrophysiology:** AAV-injected HCN1*^f/f^* mice were fully anesthetized with isoflurane and decapitated into ice-cold (4°C) artificial cerebrospinal fluid (ACSF) containing the following (in mm): 124 NaCl, 2.5 KCl, 26 NaHCO_3_, 1.25 NaH_2_PO_4_, 2 CaCl_2_, 2 MgSO_4_, and 10 glucose. Brains were dissected and sectioned in cold ACSF using a VT1200 vibrating microtome (Leica Biosystems) into coronal slices (300 μm) that contained the medial PFC. Slices were then incubated at 32 °C in oxygenated (bubbled with 95% O_2_-5% CO_2_) ACSF for 30–45 min and were moved to room temperature (22 °C) at least 45 min before the recordings began. Brain tissue was visualized under an upright light microscope (BX51WI, Olympus, Tokyo, Japan) coupled to a camera (C8484, Hamamatsu, Japan), and the cells were visualized using infrared and differential interference contrast microscopy. GFP labeled neurons in layer 5 of the prelimbic subregion and were visualized and targeted using a mercury lamp and a GFP filter set. Pipettes (open tip resistance 3–6 mΩ) were prepared from borosilicate glass (World Precision Instruments, Sarosota, FL, USA) using a pipette puller (Sutter Instruments, Novato, CA, USA) and filled with a K gluconate-based intracellular pipette solution (in mM as follows: 135 K gluconate, 5 KCl, 10 HEPES, 4 NaCl, 0.1 EGTA, 4 Mg-ATP, 0.3 Na-GTP, and 10 phosphocreatine). After reaching a >1 GΩ seal, the minimization of capacitive currents, and establishing whole-cell patch-clamp configuration, electrical activity was recorded using a Multiclamp 700B amplifier (Molecular Devices, San Jose, CA, USA) with Clampex 10.2 Software (Molecular Devices) and digitized using Axon Digidata at 10 kHz and low-pass filtered at 2 kHz. Measurements were collected under current-clamp and voltage-clamp configurations. In current-clamp, resting membrane potential (RMP) was measured shortly after the break-in and “sag ratio” was measured at a common voltage (−70 mV) using a −200 pA hyperpolarizing current step in which the difference between the peak negative membrane potential (peak voltage) was subtracted from the steady-state potential; this value was then divided by the difference between the baseline potential (base voltage) and the peak voltage and expressed as a percentage. The h-current (I_h_) was measured in voltage clamp by stepping from -50 mV to -150 mV and subtracting steady state current from the peak current.

**Data analyses:** For alcohol drinking experiments, occasionally bottles would leak causing some data points to be missed (2.1% of all values). Therefore, 2-bottle choice drinking data (daily weights (g), fluid weight consumed (g), dose consumed (g/kg) and alcohol, quinine, or saccharin bottle preference (%)) were analyzed using a mixed effects model with Gessier-Greenhouse correction. Post hoc analyses were performed using Dunnet’s multiple comparisons vs. control values. Electrophysiology data (RMP, voltage sag, I_h_) were analyzed using a 2-way ANOVA with a Sidak multiple comparison test. Group averages were expressed as the mean ± SEM unless otherwise noted. Graphing and statistical analyses were performed on Graphpad Prism 8.0 software.

## 3. Results

### 3.1. Hcn1 Gene Deletion Caused an Overall Increase in Alcohol Preference in Adolescence

We hypothesized that there would be an increase in alcohol consumption following *Hcn1* gene deletion, so we decided to use a 50:50 breeding strategy to generate C57BL/6J/129SVEV offspring. This mouse background is known to drink intermediate levels of alcohol compared to C57BL/6J mice and would be less likely to exhibit a ceiling effect [31]. At PND 30, HCN1*^+/+^* (*n* = 21), HCN1*^+/−^* (*n* = 23), and HCN1*^−/−^* (*n* = 15) littermates were given diluted 15% alcohol (*v/v*) and water under an every other day intermittent schedule [30] (Figure 1A). Throughout the drinking experiment, mouse weights increased across sessions (F(14,764) = 213.4, *p* < 0.0001), but did not significantly differ amongst genotypes (F(2,55) = 2.84, *p* = 0.07), although HCN1^+/−^ mice (23.2 ± 0.4 g) weighed more than HCN1*^+/+^* (22.2 ± 0.4 g) and HCN1*^−/−^* (21.8 ± 0.4 g) mice. Fluid consumption on drinking days, as measured by total mL per g mouse, was significantly different as a result of genotype (F(2,55) = 4.6, *p* = 0.014) with HCN*1^−/−^* (0.17 ± 0.007) and *Hcn1^+/−^* (0.17 ± 0.005) consuming less fluid than *Hcn1^+/+^* mice (0.19 ± 0.005). There was also a significant session by gene interaction (F(28,731) = 1.8, *p* = 0.007). Post hoc analyses revealed differences between HCN1*^−/−^* and HCN1*^+/+^* during sessions 4, 5 and 12 and between HCN1*^+/−^* and HCN1*^+/+^* during sessions 5, 10, 12 and 14 (Figure 1D). On the intervening days where only water was consumed, there was a main effect of genotype (F(2,56) = 6.4, *p* = 0.003) where HCN1*^+/+^* mice drank more fluid (0.17 ± 0.003) than HCN1*^+/−^* (0.17 ± 0.004) or HCN1*^−/−^* (0.15 ± 0.005) mice. There was also a significant session by gene interaction (F(28,744) = 1.62, *p* = 0.023) where post hoc analyses revealed a significant reduction in drinking on sessions 4, 6, 8 and 10 for HCN1*^−/−^* and session 10 for HCN1*^+/−^* mice compared to HCN1*^+/+^* mice. Alcohol dose consumed (g/kg) increased by session (F(7.436, 404.2) = 12.7), but did not differ by genotype (F(2,56) = 0.55, *p* = 0.58) and without any interaction (F(28,761) = 1.23, *p* = 0.19). (Figure 1B). For alcohol preference (% alcohol consumed/total fluid), a main effect of session (F(14, 738) = 19.24, *p* < 0.0001 and genotype (F(2,56) = 3.33, *p* = 0.04) were observed, but no interaction (F(28,738) = 1.32, *p* = 0.13) (Figure 1C). 

Intermittent alcohol consumption can be described as having two phases: an escalation (or initiation) phase and a maintenance phase [30,32]. Therefore, we subdivided alcohol sessions into equal epochs of alcohol escalation (sessions 1–5) and maintenance (sessions 6–10, 11–15) and ran additional analyses on the dose consumed and preference. For the dose consumed during the escalation phase, we found a main effect of session F(3.44,195.9) = 29.27 where the dose consumed increased across sessions overall, with no effect of genotype F(2,58) = 1.55, *p* = 0.22, and a significant interaction (F(8,228) = 2.11, *p* = 0.036). Post hoc analyses (Dunnet’s) revealed that the dose consumed was significantly elevated (*p* = 0.004) on day 2 in HCN1*^−/−^* mice (12.8 ± 1.2 g/kg) compared to HCN1*^+/+^* (6.8 ± 1.3 g/kg). For maintenance epoch 1 (sessions 6–10), there was a main effect of session where the overall drinking slightly decreased, but there was no effect of genotype or interaction on the dose consumed. Similarly, in maintenance epoch 2 (sessions 11–15), there was a main effect of session where the alcohol consumption was slightly decreased, but without an effect of genotype or interaction. For alcohol preference during the escalation phase, we found a main effect of session F(3.209,170.0) = 32.15 where alcohol preference increased over sessions, no effect of genotype and a significant interaction F(8,213) = 2.72, *p* = 0.007. Post hoc analyses (Dunnet’s) revealed that HCN1*^−/−^* had a significantly higher preference (%) than HCN1*^+/+^* in sessions 2 (74.4 ± 7.6 compared to 34 ± 7.3) and 5 (79.7 ± 6.7 compared to. 50 ± 7.3). For maintenance epoch 1, there was no main effect of session or genotype or interaction. For maintenance epoch 2, there was a main effect of session where preference slightly decreased overall, and a main effect of gene were HCN1*^−/−^* mice maintained a higher overall preference. Collectively, these further analyses revealed that HCN1*^−/−^* mice have a more rapid escalation of both alcohol consumption and preference not observed in HCN1*^+/+^* mice and HCN1*^−/−^* mice maintain their high preference throughout the 30 days. Interestingly, HCN1*^+/−^* tended to have intermediate values on dose consumed and preference during the escalation phase which may indicate a dose effect of gene, however, they did not reach significance in post hoc analyses. In addition, the finding that the dose consumed and preference were equal or slightly decreased during maintenance epochs supported that these maintenance and escalation phases were fundamentally different.

### 3.2. Hcn1 Gene Deletion Does Not Affect Sweet or Bitter Taste Sensitivity 

Ethanol has a bitter and sweet taste profile and HCN1 channels have been found to express in the tongue and to mediate sour taste [33]. To rule out potential confounds of bitter and sweet taste sensitivity, as well as deficits in preference and avoidance behavior, increasing concentrations of saccharine (0.033, 0.066% (*w/v*)) and quinine hemisulfate (30, 60 and 90 µM) were given as a two-bottle choice procedure with a water bottle. For saccharine, we found that there was a main effect of preference for concentration (F(1,43) = 11.1, *p* = 0.002), but not gene (F(2,44) = 2.9, *p* = 0.07) although there was a trend for HCN1*^−/−^* to have a reduced preference for saccharine (Figure 1D). Similarly, there was a main effect of quinine concentration (F(1.75,29.7) = 10.39, *p* = 0.0006) where the highest concentration had the highest avoidance, but no effect of genotype preference for quinine solution (F(1.62,27.5) = 2.03, *p* = 0.16) or concentration x genotype interaction (F(2.9,35.1) = 1.64, *p* = 0.2) (Figure 1E). These data support our finding that changes in alcohol preference were not due to sensitivity to taste. Ethanol taste sensitivity or metabolism was not directly measured in HCN1 KO mice. 

### 3.3. Viral Deletion of *HCN1* Was Effective in the Prefrontal Cortex 

At ~ PND 25, HCN1*^f/f^* mice were infused with either AAV8-CaMK2a-EFGP or AAV8-CaMK2a-CreGFP into one side of their medial PFC. Tissue was collected at 1, 2 weeks and 4 weeks after viral infusion and processed for either immunohistochemistry or whole-cell patch-clamp slice electrophysiology (Figure 2 and Figure 3). Images collected from the immunohistochemistry revealed the presence of GFP and Cre recombinase at an early (1 week) and later (2 weeks) point, and the expression of *Hcn1* in layer 1 (coinciding with dendritic expression) of either a non-injected side at 1 week (Figure 2A) or AAV8-CaMK2a-EGFP injected side at 2 weeks (Figure 2B). *Hcn1* expression was not observed in layer 1 in mice that were injected with AAV8-CaMK2a-CreGFP indicating a qualitative reduction in *Hcn1* expression that appeared to begin at 1 week (Figure 2C,D). 

From our electrophysiology recordings in GFP-positive layer 5 neurons (Figure 3A,D) of the prelimbic region of the mPFC, we observed significant differences between the neurons expressing CreGFP and EGFP control along the time points of 1 week (*n* = 9 and 5 cells), 2 weeks (*n* = 9 and 7 cells), and 4 weeks (*n* = 6 and 7 cells) in measurements of RMP (F(1,38) = 8.96, *p* = 0.005), voltage sag (F(1,34) = 15.03, *p* = 0.0005), and I_h_ (F(1,34) = 20.5, *p* < 0.0001) (Figure 3B,C,E). Post hoc analyses (Sidak’s) on electrophysiology measurements revealed a significant reduction in voltage sag (*p* < 0.05) and I_h_ (*p* < 0.005) at 2 weeks. 

From these confirmation studies, we observed that at 1 and 2 weeks, there appeared to be a reduction in the expression of *Hcn1*. At 2 weeks (but not 1 week), we saw a functional decrease in measurements of I_h_ from somatic recordings. This time period would correspond approximately to PND 32 to 40 and indicate a gradual reduction in HCN function during sessions 1–5 (escalation phase) of the HCN1 KO data. Therefore, if a gradual reduction in *Hcn1* expression in the medial PFC specifically mediates the effect seen in HCN1^−/−^ mice, we would expect a similar effect to be seen from mPFC KO behavioral data during these time points.

### 3.4. PFC Viral KO of *HCN1* Does Not Regulate Intermittent Alcohol Consumption

To test the role of HCN1 channels of the medial PFC in alcohol consumption during adolescence, HCN1*^f/f^* mice we either infused with AAV8-CaMK2a-EGFP (*n* = 13) or AAV8-CaMK2a-CreGFP (*n* = 10) bilaterally into their medial PFC at ~PND 25 and then given 5 days for recovery and viral transduction before the two-bottle choice testing. Weight and fluid consumed were not significantly different between groups. For the dose consumed (g/kg) (Figure 4A), there was a main effect of session (F(4.95,100.8) = 9.17, *p* < 0.001) where mice escalated their intake over the 15 sessions, but no effect of genotype (F(1,21) = 0.77, *p* = 0.39) or session by genotype interaction (F(14,285) = 0.74, *p* = 0.73) was observed. For alcohol preference (Figure 4B), there was a main effect for the session where preference increased over 15 sessions (F(5.16,104.3) = 8.06, *p* < 0.0001), but no main effect of genotype F(1,21) = 0.65, *p* = 0.43) or session by genotype interaction (F(14,283) = 0.45, *p* = 95). Since we observed significant effects in the HCN1 KO mice during the escalation phase (days 1–5), we ran additional analyses on this time point. For the dose consumed, we did not reveal a main effect of genotype (F(1,21) = 0.64, *p* = 0.43) or an interaction (F(4,82) = 1.32, *p* = 0.27). Similarly, comparisons of alcohol preference did not result in a main effect of genotype (F(1,21) = 0.47, *p* = 0.5) or session by genotype interaction (F(4,82) = 0.45, *p* = 0.77). Therefore, under these conditions, the mPFC deletion of HCN1 does not affect IA EtOH escalation or overall consumption and preference during adolescence.

## 4. Discussion

In this study, we found that the expression of the *Hcn1* gene constrains alcohol preference during adolescence. Male mice with a constitutive knockout of the *Hcn1* gene showed a higher alcohol preference under an intermittent schedule than their wild-type litter mates. This was most prominent during the escalation phase of alcohol drinking, indicating that a loss of HCN1 channels permits a rapid preference for alcohol that is sustained into early adulthood, whereas wild-type mice gradually reach higher preference after repeated alcohol exposures. This did not appear to be due to a loss of sweet and bitter taste sensitivity in the KO mice tested during adulthood, unless it is specific to early adolescence during the escalation phase. This result appears to be consistent with our previous finding that mice exposed to an IA EtOH procedure during adolescence have a reduction in Hcn channel function in medial PFC pyramidal neurons, escalating their alcohol preference and showing increased front-loading behavior in adulthood [21]. On the contrary, when we specifically targeted the deletion of *HCN1* channels in medial PFC pyramidal neurons, we did not observe any effect on alcohol preference or the dose consumed at time points where we confirmed a loss of *Hcn1* expression. 

While there is relative agreement that the prefrontal cortex dysfunction is likely a consequence of chronic alcohol exposure, the role of the medial PFC in alcohol consumption is only beginning to emerge. Evidence from in vivo recording studies in rodents offers some insight. In low drinking Wistar rats, population and single unit electrical recordings support the idea that neural activity in the medial PFC is associated with alcohol choice, yet in alcohol-preferring rats, this activity is significantly reduced [34]. In vivo calcium recordings performed in the mPFC of mice have found increased activity in mPFC neurons that project to the periaqueductal gray in low alcohol drinkers and a reduction in the activity in this circuit in high and compulsive alcohol drinkers during alcohol licks. Optogenetic stimulation or silencing the activity of these neurons can bidirectionally modulate compulsive alcohol intake [35]. Thus, it appears that high activity in the mPFC may serve as a set of brakes to deter maladaptive alcohol consumption. It is possible that in mPFC HCN1 deletion mice, increasing the negative consequences associated with alcohol consumption or measuring alcohol consumption after inducing dependence [36] would have more sensitivity in identifying a the role for mPFC HCN1 channels in alcohol consumption. 

The deletion of HCN1 in the medial PFC has paradoxical effects on neuronal activity. For instance, it causes effects associated with increased excitability like increased input resistance and synaptic integration by increasing the time constant, as well as the effects associated with decreased activity like resting membrane hyperpolarization and a loss of intrinsic persistent activity [28]. As such, how HCN1 loss in the mPFC would manifest in vivo during drinking is unknown. Further complicating the matter are well described differences of I_h_ magnitude in PFC neuronal populations that project to the cortical and subcortical subregions [37]. These pyramidal neuron subpopulations were not distinguishes, nor did we target the *Hcn1* gene in interneuron subtypes in the PFC [25] including Martinotti neurons which are targets of chronic alcohol [38]. Targeted silencing or the specific deletion of HCN1 in these various populations during alcohol consumption could delineate their contribution to alcohol consumption.

A major limitation of the current study is that the AAV system lacks ideal specificity. For example, it is difficult to achieve a uniform deletion of the gene in the medial PFC during early adolescence, when we observed alcohol escalation. As there are opposing roles for medial PFC prelimbic and infralimbic subregions in reward seeking and avoidance [39], it is unclear if the volume of HCN1 deletion that spans prelimbic into parts of infralimbic and anterior cingulate subregions in this study is masking the contribution of HCN1 channels from the prelimbic subregion alone. Furthermore, delineating specific populations of mPFC neurons, like pyramidal tract (PT) vs. intratelencephic tract (IT) pyramidal neurons or GABAergic interneurons, was not possible with the current strategy, so we cannot conclusively say that *Hcn1* gene deletion in the mPFC does not affect alcohol consumption. A better resolution of gene deletion in neuronal populations could be achieved using multiple virus intersect strategies [40] or BAC transgenic mouse lines [41]. As an alternative explanation, the effect seen in full animal HCN1 KO mice could very well be mediated by another brain region. Other candidate brain regions known to express *Hcn1* have a clear role in alcohol drinking, which include the orbitofrontal cortex [36] and basolateral amygdala [42]. Another major limitation is that sex was not used as a biological variable in this study but should be in the future. Female mice are known to drink more in the IA EtOH procedure [31] and display divergent adaptations to chronic alcohol [43] and female rodents have higher *Hcn1* expression in the medial PFC [37].

Previously, the *Hcn1* gene has not been associated with alcohol use, but there has been some evidence indicating it has a role in psychiatric disorders. In the amygdala, the *Hcn1* gene has been identified as a quantitative trait locus for fear learning [44]. Genome-wide association studies (GWAS) have not previously implicated HCN1 in alcohol use disorders, but other GWAS studies have found that HCN1 is associated conditions like schizophrenia and depression, which are known to be comorbid with AUDs [45]. Medications directed at the HCN channel modulation are currently being developed for seizure activity [46] and may prove useful in treating symptoms associated with AUDs associated with withdrawal-related excitability. 

## 5. Conclusions

Escalated adolescent alcohol consumption results in a reduction of HCN channel function in the PFC, suggesting that these channels may regulate heavy alcohol drinking [21]. When tested on intermittent alcohol consumption, mice with a full KO of the *Hcn1* gene showed a higher preference for alcohol than wild-type controls without lasting effects on taste sensitivity. This does not appear to be due to a gradual loss of the *Hcn1* gene in pyramidal neurons of the medial PFC during adolescence. This leaves open the possibility that the reduction in HCN function observed in the PFC after adolescent alcohol could play a role in subsequent learning and memory deficits, consistent with previous work, while changes in HCN function in other brain regions or medial PFC cell types may contribute to the higher alcohol preference observed during adolescent alcohol consumption.

## Figures and Tables

**Figure 1 brainsci-10-00763-f001:**
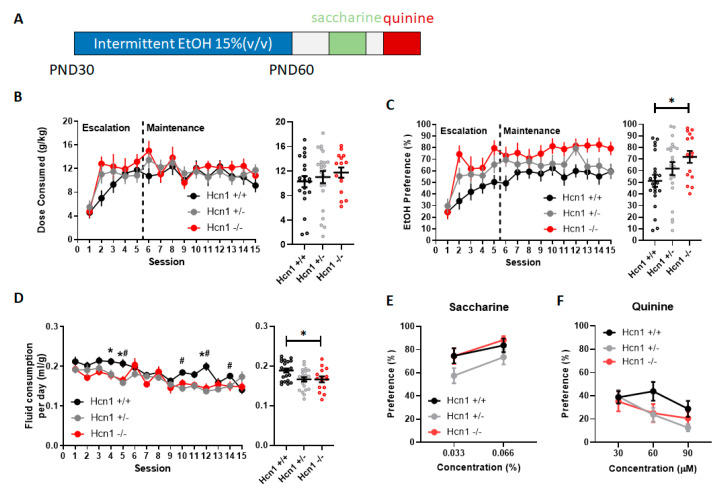
**Genetic deletion of *Hcn1* increased alcohol preference without affecting taste sensitivity.** (**A**) Experimental timeline describing the ages of mice and order of experiments. (**B**) Left, over the 15 intermittent access to alcohol (IA EtOH) sessions, alcohol dose consumed was not significantly different among genotypes HCN1*^+/+^* (*n* = 21), HCN1*^+/−^* (*n* = 23), and HCN1*^−/−^* (*n* = 15). Right, collapsed data across session. (**C)** Left, throughout the 15 IA EtOH sessions, HCN1*^−/−^* mice showed a greater preference for alcohol than HCN1*^+/+^* mice. Right, collapsed data across session demonstrate a near 20% increase in preference for alcohol throughout sessions (* *p* < 0.05) (**D**) Left, fluid consumption per day (expressed as ml per g of mouse) was significantly different across genotypes. Right collapsed data across sessions. HCN1*^−/−^* and HCN1*^+/−^* mice drank less fluid per day than HCN1*^+/+^* mice on multiple days (* *p* < 0.05, HCN1*^−/−^* vs. HCN1*^+/+^*, # *p* < 0.05, HCN1*^+−/−^* vs. HCN1*^+/+^*). To assess whether increase in alcohol preference is due to altered taste sensitivity, the mice were tested for saccharine and quinine preference. (**E**) Saccharine preference was not affected by *Hcn1* gene deletion. (**F**) Quinine avoidance was also not affected by *Hcn1* gene deletion.

**Figure 2 brainsci-10-00763-f002:**
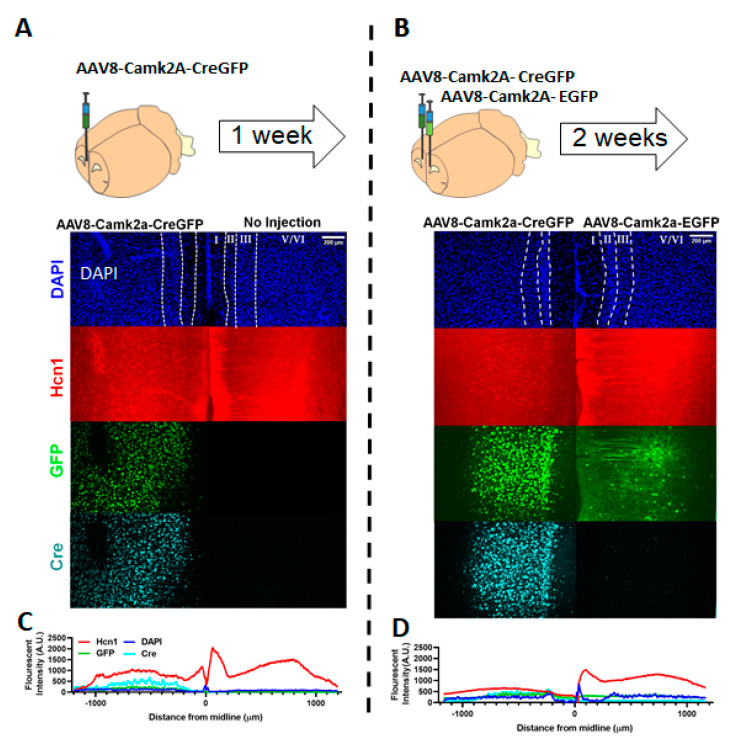
**Viral genetic deletion of *Hcn1* in medial prefrontal cortex (PFC) results in the reduction of *Hcn1* expression.** (**A,B**). Top, expression and time course of viral strategy was assessed in PFC hemispheres. (**A**) Middle panels, nuclear DAPI (4’6-diamidino 2-phenylindole) staining reveals layers of the prelimbic region (white dotted line). After 1 week of unilateral AAV8-CaMK2a-CreGFP infusion in male HCN1*^f/f^* mice, immunohistochemistry revealed the presence of GFP and Cre expression in the target hemisphere. (**C**) Average fluorescence intensity was measured across layers (A.U. arbitrary units) and the *Hcn1* expression appeared to be reduced particularly in layer 1 of the CreGFP injected side compared to the noninjected side. (**B,D**) A stronger reduction in *Hcn1* expression was apparent across layers at 2 weeks in a mouse infused with the control virus: AAV8-CaMK2a-CreGFP (left), compared to AAV8-CaMK2a-EGFP infused in the same brain (right).

**Figure 3 brainsci-10-00763-f003:**
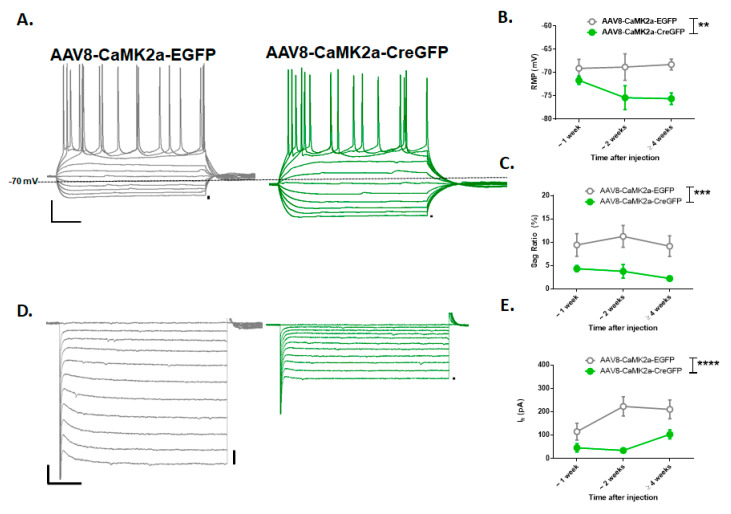
**Viral genetic deletion of *Hcn1* in PFC results in reduction in resting membrane potential (RMP), voltage sag and I_h_**. Whole-cell patch-clamp recordings were taken from the prelimbic layer 5 GFP expressing cells in the AAV8-CaMK2a-EGFP or AAV8-CaMK2a-CreGFP hemispheres. (**A**) Current clamp representative traces taken from EGFP (left, grey) or CreGFP (right, green) neurons after 2 weeks of viral injection demonstrating the characteristic pyramidal neuron firing and voltage sag after 40 pA current injection steps from −200 pA to 200 pA (x-y scale: 100 msec × 10 mV, dotted line = −70 mV). (**B,C**) CreGFP neurons were found to have hyperpolarized RMP and a reduction in sag ratio compared to the EGFP neurons consistent with a loss of HCN1. (**D**) Representative voltage clamp demonstrating a slow inward current (I_h_) observed when the neuron is stepped from −50 mV to −150 mV (x–y scale: 100 msec × 200 pA). (**E**) I_h_ magnitude is greater in EGFP neurons (grey) compared to CreGFP neurons (green) across the adolescent period. (**, *p* < 0.01, ***, *p* < 0.005, ****, *p* < 0.001).

**Figure 4 brainsci-10-00763-f004:**
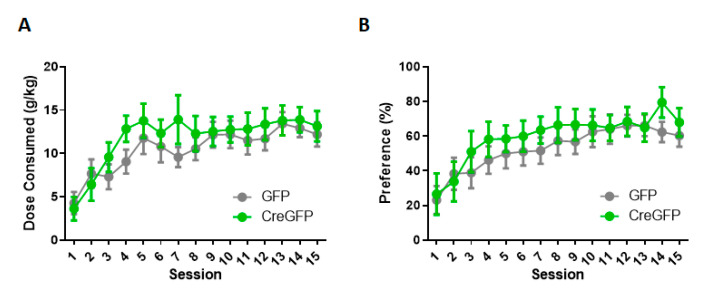
**Genetic deletion of *Hcn1* in PFC does not affect IA EtOH consumption during adolescence.** Mice bilaterally infused with AAV8-CaMK2a-CreGFP or control virus AAV8-CaMK2a-EGFP voluntarily consumed alcohol from postnatal day (PND) 30 to PND 60 on an IA EtOH 2BC schedule. (**A**) The dose consumed (g/kg) did not significantly differ between groups and neither did (**B**) the preference for the alcohol bottle over the water bottle.

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
