# Peer review of "Constitutive Genetic Deletion of *Hcn1* Increases Alcohol Preference during Adolescence"

_brainsci, 2020, doi:10.3390/brainsci10110763_

Round 1

Reviewer 1 Report

This article examines the effect of global and focal (medial prefrontal cortex; PFC) deletion of the Hcn1 gene on voluntary ethanol consumption and preference. The manuscript adequately tests their hypothesis that HCN1 knockout, particularly in mPFC, elevates voluntary alcohol consumption, that is not the product of altered taste reactivity. Their observations demonstrate some role for HCN1 in the acquisition of alcohol preference, but refute any specific role for mPFC-expressed HCN1 in this phenomenon. While of interest to the field, a number of concerns should be addressed prior to publication of the manuscript.

  1. That EtOH preference in Hcn1 -/- mice is consistently elevated with no consistent effect on consumption is puzzling and limits the significance of the result, particularly since the authors note that homozygous null mice display significantly reduced liquid intake overall. Is that the more biologically relevant observation?
  2. Did the authors evaluate drinking microstructure, or evaluate potentially different ethanol metabolism or elimination profiles affecting drinking behavior? This could also dramatically alter the interpretation of the results.
  3. The authors should provide some justification for not using female mice in this study, particularly since female rats display much higher levels of HCN1 protein in mPFC relative to males (Hughes et al., 2020).
  4. Please report statistics for the main effect of quinine concentration, as the figure is not particularly convincing.
  5. Please include scale bars or laminar outlines (or both) in the IHC panels.
  6. While the authors note in the discussion that other cortical neurons express HCN1, they fail to mention that the presence of a robust HCN-mediated voltage sag is the defining characteristic of Martinotti interneurons, a prominent source of intra-cortical feedback inhibition (Silverberg, 2007).
  7. There are a number of grammatical errors in the manuscript that should be addressed (e.g. Lines 161, 225, 229, 269).

Hughes BA, Crofton EJ, O’Buckley TK, Herman MA, Morrow AL (2020). Chronic ethanol exposure alters prelimbic prefrontal cortical fast-spiking and Martinotti interneuron function with differential sex specificity in rat brain. Neuropharmacology 162 

Silberberg G, Markram H (2007). Disynaptic inhibition between neocortical pyramidal cells mediated by Martinotti cells. Neuron 53(5): 735-746.

Reviewer 2 Report

In this study, Salling and Harrison explored the role of cortical HCN1 channels in alcohol drinking using adolescent constitutive Hcn1 knockout mice as well as adolescent conditional Hcn1 knockout mice using cre-recombinase driven expression in Camkii-expressing mPFC neurons. They find that constitutive knockout transiently increases consumption of 15% alcohol and lasting increases in alcohol preference in an intermittent 2 bottle choice drinking. In their mPFC-specific knockout they find no effect of genetic knockout. The subject is very important and the premise of their studies is sound and quite exciting. While I am not overly skeptical of their overall findings, some the data presented are not as convincing as I would have hoped and there are some limitations of their study design that complicate the interpretation of their data.

Major concerns:

  1. The data in Figure 2 are not sufficiently convincing as presented. I would be surprised if their cre- vectors didn’t knockout Hcn1, but what is there is not sufficient for a variety of reasons.
    1. The immunostaining is very poor. To my eyes I cannot see the knockout of HCN1 that they are say is there. Some ways to solve this problem are to 1) Show much higher magnification images; 2) perform HCN1 immunostaining in the constitutive knockout to show antibody specificity; 3) provide images of their 2 weeks as well.
    2. It is possible that their electrophysiology data are correct but there are some critical data not shown and caveats that have not been addressed. They need to show representative traces for all recordings. They do not consider that HCN1-mediated Ih is a characteristic of PT pyramidal neurons (vs. IT) but the authors do not demonstrate that they limited their recordings to PT neurons so the comparisons between vector treatments are not necessarily an equal comparison. This could influence the rate at which the functional changes occurred (i.e. the 1 week effect may not actually have reduced function). They imply, but do not explicitly state, that this is a problem in their discussion (ref 32, lines 264-265).
    3. They did not report from which layer of cortex they performed their recordings.
  2. The authors clearly were looking at a gene-dose effect in Figure 1 with the heterozygotes, but these mice were barely mentioned or discussed throughout the paper. Many times, the statistical reporting did not even state anything about comparisons to these mice. The legend of Figure 1C specifically showed the Hcn1 -/- was different than the WT, but the panel made it look like it was different both the WT and +/-. The Figure legend indicated what the different *’s represented, but since there were 3 genotypes it was not clear on the individual days what the difference was. In Figure 1D there is clearly a difference with the +/-, but the text said it was the -/- which is either a typo or a problem with the figure.
  3. The time courses of Figure 1B and 1C seem to suggest that the impact of a loss of Hcn1 expression may be most robust in the initiation of drinking rather than maintenance. The lasting increase in preference is difficult to disambiguate whether this is a persistent effect from the initiation or is part of maintenance. Having the total fluid consumption data presented in graphical format would be useful here as well. The specific problem here is that in Figure 3 where they find no effect of the mPFC knockout, they cannot be sure that Hcn1 is fully knocked out during the initiation stage (see Figure 2). Therefore if HCN1 is specifically involved in initiation, but it is not fully gone during the initiation, they may be getting a false negative for a role for mPFC HCN1 in regulating alcohol consumption. I recognize this is a limitation of the study due to their focus on adolescent drinking, so this point at the very least should be explicitly discussed in detail.
  4. Why were multiple concentrations of sucrose and quinine used, but not for alcohol which is more relevant to the overall message of the paper? There should be a dose-response analysis here.
  5. Were there motor effects related to Hcn1 knockout? This needs to be reported.

Minor concerns

  1. There were many typos throughout the manuscript. The authors should perform a more careful proofread.
    1. Some examples: media (line 122), groups() (line 225), p=95 (Line 229), Hcn function (should be Hcn1, line 286).
  2. Are the constitutive knockout mice truly transgenic (i.e. was the knockout produced via an insertion of a transgene)? If not, they should not technically be referred to as transgenic mice.
  3. The way the sentence from lines 134-139 was awkwardly written.
  4. There was no explanation for why RM Mixed factor ANOVAs were used in for alcohol data but not for quinine or saccharine.
  5. Body weights were reported to be lower overall in the KO mice, but the data were not shown.
  6. Figure 1B was reported to only have one day of significant difference, but the figure shows two.
  7. A supporting reference for the first sentence of the conclusion should be provided.
  8. There were some statements that seemed odd to see in a finalized version of a manuscript: lines 294-295, 307-308.

Reviewer 3 Report

This manuscript by Salling and Harrison describes manipulation of Hcn1 channels globally and selectively in medial prefrontal cortex pyramidal neurons of adolescent mice drinking in the intermittent alcohol model. The major finding of the manuscript is that global deletion, but not local deletion of Hcn1 decreases preference for, but not consumption of alcohol. The findings are interesting and the experiments are well-designed as a follow-up to their recent J Neuroscience manuscript on h-current in the medial PFC. There are a few concerns that should be addressed that might improve the manuscript.

(1) The statistics for the drinking and preference study are not reported correctly. Since the interactions were not significant, the post-hoc tests for session-by-genotype are invalid. Rather, the authors should report the post-hocs for session collapsed across genotype and genotype collapsed across session. To highlight the main effect of genotype on alcohol preference, the authors should consider adding a graph of averaged preference across the entire 15 drinking sessions for each of the three genotypes. The results of these post-hoc analyses will tell us which genotypes are different. Accordingly, the asterisks shown in Fig 1B and Fig 1C for individual drinking sessions should be removed.

(2) The authors should consider revising the interpretation of the results given that the statistics do not support a role for mPFC Hcn1 channels in pyramidal neurons as mediators of excessive alcohol drinking (e.g., see lines 243-245 and line 283).

(3) It would be great if the authors could discuss how their findings on Hcn1 in mPFC pyramidal projection neurons integrate into the existing literature, especially with regard to lines 249-259. The mPFC can control dependence-induced increases in drinking or compulsive intake. Is it possible that Hcn1 channels in mPFC could regulate drinking under these conditions rather than drinking in the IA model?

Minor comments:

  • Scale bars and representative traces should be added to Fig 2.
  • There are many errors that should be corrected (lines 193, 225, 269, among others).
  • F values are missing for some of the statistical tests, including line 225.
  • Based on the main effects and lack of interactions, the text on lines 203 and 204 should be adjusted.
  • There are recent reviews on acute and chronic effects of alcohol on intrinsic excitability that could be referenced for the sentence on line 41.
  • Typically, gene names are shown in italics.
  • The authors should consider adding a statement describing the mPFC subregion(s) where Hcn1 was manipulated. If Hcn1 expression was decreased in multiple subregions, could this explain the negative findings given that these subregions sometimes have opposing roles in drug-seeking behaviors?

Round 2

Reviewer 1 Report

The authors have made some beneficial revisions.  Indeed, they agree with the caveats raised by the reviewers, but don’t change their title or conclusions in the paper. They didn’t add justification for study of only male rats and didn’t cite the difference in HCN in female vs male rats, either.  They find the biggest effect on behavior (escalation of drinking) when the KO is incomplete - before two weeks.  After two weeks when the KO is clear, the behavioral adaptation is questionable and mostly due to changes in all fluid consumption.  The title does not reflect this reality. 

Author Response

Uploaded word file

Reviewer 2 Report

The authors have satisfied my concerns. I have a few editorial concerns that I should be addressed:

Figure 1D is lacking a description in the figure legend and the units are odd. Fluid consumption should be normalized to the actual body weight as in other figure panels (e.g. g fluid/kg).

Missing word in Figure 1 legend: “When escalation of dose consumed was specifically analyzed, a significant session x genotype _____ was revealed.”

Acknowledgements still indicate HCN1 mice as "transgenic."

Author Response

Uploaded as word file
